# New Therapeutic Strategies for Obesity and Its Metabolic Sequelae: Brazilian Cerrado as a Unique Biome

**DOI:** 10.3390/ijms242115588

**Published:** 2023-10-25

**Authors:** Tamaeh Monteiro-Alfredo, Maria Lígia Rodrigues Macedo, Kely de Picoli Souza, Paulo Matafome

**Affiliations:** 1Coimbra Institute of Clinical and Biomedical Research (iCBR), Faculty of Medicine, University of Coimbra, 3000-548 Coimbra, Portugal; tamaehamonteiro@hotmail.com; 2Center for Innovative Biomedicine and Biotechnology (CIBB), University of Coimbra, 3000-548 Coimbra, Portugal; 3Clinical Academic Center of Coimbra, 3000-075 Coimbra, Portugal; 4Research Group on Biotechnology and Bioprospection Applied to Metabolism and Cancer (GEBBAM), Federal University of Grande Dourados, Dourados 79804-970, MS, Brazil; kelypicoli@gmail.com; 5Laboratório de Purificação de Proteínas e Suas Funções Biológicas (LPPFB), Federal University of Mato Grosso do Sul, Campo Grande 79070-900, MS, Brazil; ligiamacedo18@gmail.com; 6Coimbra Health School (ESTeSC), Polytechnic University of Coimbra, Rua 5 de Outubro, 3046-854 Coimbra, Portugal

**Keywords:** phytochemical compounds, secondary metabolites, metabolic diseases, ethnopharmacology, Cerrado ecosystem

## Abstract

Brazil has several important biomes holding impressive fauna and flora biodiversity. Cerrado being one of the richest ones and a significant area in the search for new plant-based products, such as foods, cosmetics, and medicines. The therapeutic potential of Cerrado plants has been described by several studies associating ethnopharmacological knowledge with phytochemical compounds and therapeutic effects. Based on this wide range of options, the Brazilian population has been using these medicinal plants (MP) for centuries for the treatment of various health conditions. Among these, we highlight metabolic diseases, namely obesity and its metabolic alterations from metabolic syndrome to later stages such as type 2 diabetes (T2D). Several studies have shown that adipose tissue (AT) dysfunction leads to proinflammatory cytokine secretion and impaired free fatty acid (FFA) oxidation and oxidative status, creating the basis for insulin resistance and glucose dysmetabolism. In this scenario, the great Brazilian biodiversity and a wide variety of phytochemical compounds make it an important candidate for the identification of pharmacological strategies for the treatment of these conditions. This review aimed to analyze and summarize the current literature on plants from the Brazilian Cerrado that have therapeutic activity against obesity and its metabolic conditions, reducing inflammation and oxidative stress.

## 1. Introduction

In the last decades, there has been a significant increase in the life expectancy of the global population. According to the World Health Organization (WHO), between the years 2000 and 2015, life expectancy increased by an average of 5 years [1]. This remarkable improvement can be attributed to several factors, including a better quality of life, improved access to healthcare services, and the implementation of public policies aimed at disease prevention and treatment [1]. Although higher life expectancy is often associated with improved quality of life, it has also led to a higher incidence of chronic diseases worldwide, creating a significant burden on healthcare systems [2,3,4], requiring the development of more effective and affordable therapies to address this growing challenge.

Among the chronic illnesses that have been progressively more common lately, we highlight obesity, which is usually caused by unhealthy lifestyle habits—such as the consumption of processed foods rich in long-chain saturated fatty acids and added sugars [5], a sedentary lifestyle, and overweight [6]—and affects the overall quality of life of the population, both physically and economically, as well as psychologically [7]. Obesity is considered a major public health problem and is ranked as the fifth leading cause of mortality worldwide [8]. In addition to potentially leading to death, it can contribute to the development of other chronic disorders, including cancer [9], respiratory, hepatic, renal, and cardiovascular diseases (CVD) [10,11], and other metabolic disorders such as metabolic syndrome [8] and type 2 diabetes (T2D) [5,12].

The World Health Organization (WHO) has established overweight as a body mass index (BMI) between 25 and 29.9 kg/m^2^, and obesity is defined as a BMI above 30 kg/m^2^ [13]. Although BMI is a widely used parameter for determining obesity, there are limitations in its definition because this index does not differentiate between lean mass and fat, nor does it evaluate body fat distribution [14]. Additionally, besides defining parameters for the diagnosis of obesity, the WHO has also predicted that the adoption of unhealthy lifestyles will be responsible for 30% of deaths by the year 2030. Even though these data are alarming, they can be reversed through the identification of risk factors and the implementation of public policies promoting healthier habits [8]. Recent research [15,16,17] has observed that obesity-associated risks are more complex than simply weight excess. These studies highlight fat distribution as a crucial factor in the development of obesity-related comorbidities. Excess visceral fat also leads to the development of pro-oxidant conditions [11], increased proinflammatory cytokines, AT dysfunction, insulin resistance, pancreatic beta-cell dysfunction, and the hepatic accumulation of free fatty acids (FFA) [17,18]. Insulin resistance itself causes abnormal glucose and lipid metabolism in adipocytes, hepatocytes, and muscle cells, leading to a systemic metabolic dysregulation [19]. This systemic dysmetabolism results in redox imbalance which can trigger a variety of consequences linked with protein oxidation and glycation, lipid peroxidation, and the release of inflammatory signaling molecules. 

In this context, it is important to highlight the importance of secondary metabolites of plants, which have therapeutic properties capable of contributing to the restoration of this imbalance in a comprehensive manner. These phytochemical compounds act through several mechanisms, such as capturing free radicals, activating or inhibiting enzymes, and specific transcriptional factors (such as Nrf2-Keap1, NF-κB, MAPK), regulating cellular activities, and also have modulatory effects on inflammatory processes (for example, in arachidonic acid metabolism, involving phospholipase A2 and COX, and in arginine metabolism, including NOS), enzymatic expression, and cytokine release [20,21]. The development of therapeutic options for obesity and its related diseases is crucial, especially if they have low cost, easy accessibility, low toxicity, and, most importantly, significant pharmacological potential. Approximately 65% to 80% of primary healthcare in developing countries depends on the consumption of medicinal plants (MP) [22], and more than 50% of the newly developed drugs are derived from them or their constituents [23]. However, MP accounts for only 15% of the existing 300,000 plant species worldwide [24]. Therefore, they offer a vast source of chemical compounds that are still underexplored and can be used in the development of new therapeutic alternatives [25].

Brazil is home to the world’s most extensive biodiversity, containing a rich variety of flora and fauna, including significant biomes such as the Cerrado [26]. This makes the country a crucial area for exploring new alternatives for producing plant-based products, including food, cosmetics, and medicines [27]. The high acceptance of plant-based treatments among the population highlights the importance of developing new treatment alternatives, especially in the field of obesity. The vast biodiversity of the Cerrado and the resulting diverse chemical composition of plants in it may reveal significant pharmacological effects. Several studies have explored this natural potential, and here we highlight the native plants of the Cerrado, which have demonstrated therapeutic effects, particularly in terms of anti-obesogenic properties, and/or their ability to assist in the treatment of its causes and consequences, such as AT inflammation, oxidative stress, CVD, and overall metabolic conditions.

## 2. Chemical Constitution of Native Plants of the Brazilian Cerrado

Brazilian biodiversity has significant economic value in the production of dietary and plant products, due to the diverse range of primary and secondary metabolites found in plants [27]. Secondary metabolites contribute to the chemical complexity of plants and are associated with their biological activities. The pharmacological potential of these plants, based on their phytochemical compounds, can aid in restoring the body’s redox balance [27] and chronic conditions [28,29]. Some metabolites that have biological effects related to the context of obesity and its complications are tannins [30,31], coumarins [32,33], terpenoids [34], anthocyanins [35,36], saponins [37], phenolic compounds, and flavonoids [38].

The efforts of science have demonstrated the increasing potential of MPs and their phytochemicals in reducing obesity. The use of multiple combinations of phytochemicals can enhance their beneficial effects at the molecular, cellular, and metabolic levels, providing advantages over treatments based on chemically synthesized drugs. Moreover, toxicity is usually found only for higher dosages, increasing the security of MPs. Plants and their derived compounds have the potential to control appetite, inhibit pancreatic lipase activity, induce lipolysis, stimulate thermogenesis, lipid metabolism and adipogenesis, and regulate fat cell cycle [39]. Various compounds extracted from plants are capable of regulating the different stages of adipocyte development, becoming important therapeutic strategies in the treatment of obesity [39]. Considering the wide range of compounds that have shown therapeutic activity in the field of obesity and its complications, we will provide a more detailed description of the two major groups that have demonstrated significant therapeutic potential: polyphenols and terpenes/sterols.

### 2.1. Polyphenols

Polyphenols constitute one of the largest classes of biologically active secondary metabolites. They are represented by over 8000 compounds identified in various plants, classified based on the number of phenolic rings, molecular structure, and functional groups attached to the phenolic rings. Their two main classes are flavonoids and non-flavonoids (phenolic acids, lignans, and stilbenes) [40]. These metabolites are responsible for a significant portion of the antioxidant activity found in plants. They exert their effects by chelating metal ions through redox reactions, inhibiting the transformation of hydroperoxides into reactive oxygen species (ROS), and acting as scavengers of free radicals by donating electrons from hydroxyl groups. The potential of phenolic compounds also lies in the ability to donate hydrogens or electrons, facilitating the stabilization of intermediate radicals [2]. 

The treatment of adipocytes with polyphenolic compounds has been shown to significantly stimulate the lipolysis and β-oxidation of fatty acids, reducing the accumulation of fatty acids and proinflammatory factors and exhibiting an inhibitory effect on their differentiation [39]. This is associated with data demonstrating the effect of polyphenols and their derivatives in improving the obesity condition and lipid profile of treated animals, associated with the increased utilization of fat for energy expenditure and glycemic balance [12,41,42]. These effects are associated with the activation of specific pathways that are crucial for the metabolic balance of the organism, namely the activation of peroxisome proliferator-activated receptors (PPARs), AMP-activated protein kinase (AMPK), and glucose transporters 2 and 4, which are further detailed in the following sections [12,39].

Regarding flavonoids, they not only possess antioxidant effects by reducing ROS accumulation through the capture of free radicals but also have a wide range of health benefits in humans due to their bioactive properties [43]. These include anti-inflammatory, anticancer, anti-aging, cardioprotective, neuroprotective, immunomodulatory, antidiabetic, antibacterial, antiparasitic, antiviral [44,45], and photoprotective activities [46]. Regarding the metabolic effect of flavonoids, they have shown lipolytic and synergistic effects when combined with other compounds, beyond their cytostatic and thermogenic potential [39]. In addition to the described effects of polyphenols per se, it is also important to highlight that these bioactive compounds undergo metabolism by various enzymes in the human body, leading to structural changes that result in improved bioavailability and enhanced effects [12,47]. Although the metabolic benefits of polyphenols have been extensively evaluated in animal models of obesity and metabolic disorders, their performance in clinical trials has not been so promising, and most of the studies until now demonstrate no effect or a very modest effect. Importantly, most of the studies use one single compound, and it is possible that a therapeutic effect may only be achieved with combinations, like in MP extracts, which have a mixture of compounds and may be the mixture that has therapeutic effects.

### 2.2. Terpenes and Sterols

Some plants composed of terpenoids and their derivatives, known as sterols, have shown potential effects in the treatment of metabolic syndrome and its complications. Terpenes are composed of isoprene units with five carbons and one carbon side chain. They include carotenoids such as β-carotene, a precursor of vitamin A [48], ursolic acid [49], tetraterpenes [50], and pentacyclic terpens [51,52], among others. The class of sterols includes sphytosterols as campesterol, sitosterol, and stigmasterol, which are an important part of dietary intake [48]. Regarding their biological effects, some plants containing terpenes have demonstrated antioxidant, antimicrobial, and anticancer activities. 

Terpenes can also act as anxiolytic, immunomodulatory, analgesic, anti-inflammatory, neuroprotective, hepatoprotective, and wound-healing agents. They have also been studied for their potential against metabolic disorders, particularly obesity and T2D [53]. Specifically, concerning metabolic disorders, terpenes can activate PPARs, known for their anti-inflammatory action [34] (discussed in more detail in the section about inflammation and obesity).

In the article selection stage for this review, the scientific databases Science Direct and PubMed were used as primary sources. The keywords chosen for the search covered a variety of topics related to obesity and the Cerrado biome. Priority was given to articles that contained information about native plants, with a special focus on phytochemical composition. The inclusion of articles that met these criteria was carried out with the aim of adequately covering the intersection of topics of interest in this review. In summary, in Table 1 we highlight studies that link the chemical composition of plants to their various therapeutic effects, particularly regarding their potential for metabolic improvement, anti-obesogenic effects, and addressing of the causes and consequences of obesity, such as oxidative stress, inflammation, diabetes, and CVD. In addition, we summarized these data in Figure 1, correlating the therapeutic effect of the plants with the chemical constitution. These data support the importance of this biome as a source of bioactive compounds for the bioprospecting of future drugs and therapeutic strategies, given the complex chemical compositions and diverse metabolites found within it.

**Table 1 ijms-24-15588-t001:** Phytochemical constitution and biological properties related to obesity and its metabolic sequelae of medicinal plants from the Brazilian Cerrado.

Medicinal Plant	Phytochemical Constituents	Biological Properties	References
*Acrocomia aculeata*	Gallic, vanillic, caffeic, and ferulic acid, rutin, quercetin, campesterol, stigmasterol, β-sitosterol, lupeol, and lupeol acetate	In vitro and in vivo antioxidant activity, hypoglycemic, hypotriglyceridemic anticancer and cardioprotective effect	[2,12,54]
*Alibertia edulis*	Caffeic acid, quercetin 3-rhamnosyl-(1 → 6)-galactoside and iridois ioxide	Hypoglycemiant effect, protection against hemolysis and oxidative stress	[55]
*Alibertia verrucosa*	Phenolic compounds and tocopherols	Antioxidant activity	[56]
*Anacardium humile*	Phenolic compounds, anthocyanins and tocopherols	Antioxidant activity	[56]
*Annona crassiflora*	Epicatechin and quercetin	Antioxidant, antiproliferative and wound healing	[57]
	Alkaloids, specially the isolated one, stevagallin	Anti-obesity capacity, inhibition against pancreatic lipase with low cytotoxicity	[58]
*Annona muricata*	Total phenolic compounds, flavonoids and proanthocyanidins (total quantification)	Antioxidant activity, in vitro antidiabetic and inhibition of α-amylase, lipase, α-glucosidase, non-enzymatic glycation, and lipid peroxidation	[59]
	Phenols, flavonoids, saponins, tannins, steroids, and alkaloids	Antidiabetic and antiglycation	[60]
*Bactris setosa*	Phenolic compounds (anthocyanins and non-anthocyanin phenolic compounds) and carotenoids	Oxidative and nitrosative protection	[61,62]
*Banisteriopsis argyrophylla*	Catechin, flavonoids, glycosylated kaempferol, procyanidins, and megastigmane glucosides	α-amylase, α-glucosidase, lipase, and glycation inhibitors, antidiabetic and antioxidant	[63]
*Byrsonima verbascifolia*	Resveratrol and ferulic acid	Antimutagenic, antigenotoxic and antioxidant activity	[64]
*Buchenavia tomentosa*	Phenols, carotenoids and tocopherols	Antioxidant activity	[56,65]
*Campomanesia cambessedeana*	Catechin, ethyl gallate and propyl gallate	Antimutagenic, antigenotoxic and antioxidant activity	[64]
*Caryocar brasiliense*	Gallic acid, quinic acid, quercetin, and quercetin 3-O-arabinos	Antioxidant activity	[66]
	Phenolic acids and tannins, including corilagin and geraniin	Antidiabetic effect	[67]
*Cedrela odorata*	Gallic acid, catechin and gallocatechin	Hyperglycemia reduction and antioxidant activity in vivo	[68]
*Dipteryx alata*	Gallic acid and its derivatives, such as gallic acid esters and gallotannins	Antioxidant and antiproliferative activity	[69]
	p-Coumaric, ellagic, caffeic, ferulic, and gallic acid and hydroxybenzoic, catechin and epicatechin	Antioxidant activity	[70]
	Phenolic compounds (total quantification)	In vivo antioxidant activity	[71]
	Phenols and terpenes	Antioxidant activity and Caenorhabditis elegans life expectancy increase	[72]
*Eschweilera nanat*	Rutin and hyperoside	Antioxidant activity	[73]
*Eugenia dysenterica*	Proanthocyanidins, flavonoids, phenolic acids, quercetin, kaempferol derivatives, free and total ellagic acid	Antioxidant activity, pancreatic lipase inhibition, lower body weight and fat mass, improved hyperglycemia, dyslipidemia and fecal triglycerides excretion	[74]
	Quercetin and gallic acid	Inhibitory effects on hydrolases, antioxidant and antiglycation properties	[75]
	Phenolic compounds (total quantification), myricetin, quercetin and kaempferol	Antioxidant, antiproliferative and antimutagenic potential	[76]
*Eugenia klotzschiana*	More than 35 compounds (see ref.)	Antioxidant and antibacterial effect	[77]
*Guazuma ulmifolia*	Flavan-3-ol-derived flavonoids, including monomers and dimers, condensed tannins, and glycosylated flavonoids	In vitro and in vivo antioxidant activity	[10]
*Hancornia speciosa*	Phenolic compounds (total quantification)	Antioxidant activity	[78]
*Hymenaea stignocarpa*	Caffeic acid, quercetin-3-rutinoside, kaempferol; quercetin-3-rhamnoside	α-amylase and α-glucosidase inhibition, glycemic profile improved	[79]
*Hyptis Jacq.*	Phenolic acids, flavonoids, cinnamic acid derivatives, chlorogenic acid and rosmarinic acid	Antioxidant activity	[80]
*Kielmeyera coriacea*	Protocatechuic acid, procyanidins A, B, and C and epicatechin	Antioxidant and antiglycation activity, and LPL inhibition.	[81].
*Mauritia flexuosa*	Total phenolic compounds and β-carotene (total quantification)	Antioxidant activity	[82]
	Carotenoids and polyphenols (catechin, quercetin and gallic acid)	Antioxidant activity, antimutagenic, antimicrobial	[83,84,84,85,86,87,88,89]
*Mauritiella armata*	Palmitic, estearic, oleic, linoleic, linolenic acid, tocopherol, and α-tocopherol	Antioxidant activity	[90]
*Myrcia bella*	Flavonoids and phenolic acids derivatives	Antimutagenic and antioxidant activity	[91]
*Passiflora setacea*	Polyphenols	Antidiabetic and anti-inflammatory effect, with insulin, HOMA IR, PPAR-γ and IL-6 levels improvement	[92]
*Pouteria ramiflora*	Friedelin, epifriedelanol, taraxerol, triterpens and FFA	Antioxidant and α-amylase inhibition	[93]
*Pouteria torta*	Phenolic compounds, flavonoids, catechin and epicatechin	Antioxidant and α-amylase inhibition	[94]
*Psidium cattleianum*	Epicatechin, gallic, coumaric, and ferulic acid, myricetin and quercetin	Antioxidant and antimicrobial activity and antiproliferative effect on human cancer cells	[95]
*Rollinia mucosa*	Phenolic compounds and tocopherols	Antioxidant activity	[56]
*Schinus terebinthifolius Raddi*	O-glycosylated flavonols, gallotannins and gallic acid along with its derivatives	Antioxidant, antidiabetic and antiproliferative activities	[96,97]
*Sterculia striata*	Oleic acid, phytosterols β-sitosterol, stigmasteroland, campesterol γ-, δ-, α- and β-tocopherol, ellagic, ferulic, methoxyphenylacetic and protocatechuic acids	Antioxidant activity	[98]
*Solanum lycocarpu*	24 phenolic compounds (see ref.)	Antioxidant activity	[99]
*Vochysiaceae species*	Polyphenols, such as flavonoids and condensed tannins	Antioxidant and inhibitory potential against human α-amylase and protein glycation	[100]
*Senna velutina*	21 compounds (see ref.)	Antioxidant, in vitro and in vivo antitumor effects	[101,102]

It is important to underscore that several studies have assessed the toxicity of various plants, with many confirming their safe use [2,54,103,104,104,105,106,107]. These studies provide additional evidence supporting their therapeutic potential, which is the focus of this work. While some plants are often safe for use, it is crucial to acknowledge the toxic potential of plants. Due to the richness of phytochemical compounds, its metabolization can generate toxic metabolites, mainly in the liver. The literature has not yet yielded definitive information on the mechanisms responsible for liver toxicity. However, factors such as high lipophilicity or elevated doses can potentially elicit such effects [108]. Furthermore, after demonstrating the importance and chemical richness of the plants found in the Brazilian Cerrado, we present their therapeutic potential on the pathophysiological mechanisms of obesity and its metabolic complications.

## 3. Obesity, Dyslipidemia and Inflammation

Obesity is most of the time characterized by AT dysfunction, which is closely associated with obesity-related diseases, such as CVD, hypertension, metabolic syndrome, and T2D [11,40,109,110,111]. AT dysfunction encompasses adipocyte hypertrophy, leukocyte infiltration and an exacerbated secretion of proinflammatory cytokines, including tumor necrosis factor alpha (TNF-α), interleukins IL-6, IL-1β, and IL-18, and monocyte chemoattractant protein-1 (MCP-1), resulting from nuclear factor kappa B (NF-κB) activation [112]. Furthermore, the accumulation of excessive free fatty acids (FFA) in the muscles and liver contributes to the activation of the c-Jun N-terminal kinase pathway (JNK). All these pathways hinder insulin-induced tyrosine phosphorylation and PI3K-AKT activation, resulting in insulin resistance [113]. AT inflammation culminates in impaired tissue plasticity, the installation of chronic low-grade inflammation, and persistent insulin resistance [34]. Adipocyte insulin resistance also increases the level of FFA in the bloodstream due to lower lipolysis inhibition [114]. Obesity also causes inflammation in the liver, skeletal muscle, and vessels (atherosclerosis), in part due to an increased efflux of FFA from AT to these tissues [114].

Visceral adipose tissue (VAT) has a higher rate of lipolysis and macrophage infiltration than subcutaneous adipose tissue (SAT) [115], especially proinflammatory M1 macrophages [114]. However, recent studies conducted in humans have suggested a subdivision between metabolically healthy and unhealthy obesity. Metabolically healthy obese individuals show better insulin sensitivity and have lower levels of ectopic fat, improved AT function, and lower cardiometabolic risk than metabolically unhealthy obese patients [114,116,117]. However, the molecular determinants of both conditions are yet to be established. Adipose tissue may also be affected by gluco- and lipotoxicity. The accumulation of AGEs and their precursors is also associated with local inflammation, impairing microcirculation and the expansion of AT, which can result in hypoxia and insulin resistance, especially when associated with high-fat diets and obesity [18,114]. Furthermore, elevated FFA levels increase superoxide (O_2_^−^) generation, redox imbalance and lipid peroxidation [118]. Higher levels of FFA are associated with higher levels of 4-hydroxy-2-nonenal (4-HNE) in obese patients [119]. Abnormalities of mitochondrial function and biogenesis through a downregulation of PPARγ and PPAR co-activator 1-α (PGC-1α) are related to oxidative stress [120,121]. Mitochondrial dysfunction and reactive oxygen species (ROS) are associated with the dysregulation of adipokine secretion, fatty acid oxidation, and glucose homeostasis [11]. Table 2 summarizes the data found for plants that act on the regulation of inflammation, adipogenesis, browning, endocrine mechanisms, and enzyme activity.

Both crude extracts and isolated phytochemical compounds may become phytotherapeutic strategies for weight maintenance and metabolic homeostasis [115]. The main mechanisms by which phytochemical compounds from plants may improve obesity may include enzymatic inhibition, anti-inflammatory and antioxidant effects, appetite reduction, increased thermogenesis and lipid metabolism, and the regulation of adipogenesis [39,122,123]. The extract of *Casearia sylvestris,* composed mainly of the flavonoid glycosides and rutin, reduced dyslipidemia and atherogenesis caused by a high fat diet, besides improving the levels of cholesterol fractions and triglycerides, in Swiss mice [124]. The study conducted by Sousa et al. evaluated the effect of *Davilla elliptica* on diet-induced hepatic steatosis also in Swiss mice. Animals treated with the plant extract fraction and leaf powder showed reduced fat deposition in the liver, a lower body weight, and lower levels of total cholesterol and triglycerides [125].

### 3.1. Antilipidemic Effects of Cerrado Plants through Enzymatic Inhibition

Lipoprotein lipase (LPL) is responsible for metabolizing triglycerides from chylomicrons, allowing fatty acid uptake by adipocytes and triglyceride storage. Changes in LPL activity or its availability are directly related to higher concentrations of triglycerides in the blood, which can be stored in other tissues or metabolized [126]. The study conducted with *Kielmeyera coriacea* confirmed, in addition to antioxidant and antiglycation effects, the inhibitory effect on LPL, as well as other enzymes related to carbohydrate metabolism in vitro. The compounds found, including protocatechuic acid, epicatechin, and procyanidins A, B, and C, are likely responsible for the antioxidant, hyperglycemic, and hyperlipidemic effects of the plant [81].

Regarding digestive enzymes, some researchers have recently associated the inhibition of α-glucosidase and lipase with a reduction in postprandial glucose and appetite reduction, which can be positive for the treatment of obesity [127]. Many studies evaluated the inhibitory profile of phenolic compounds on digestive enzymes in vitro, including amylase, glucosidase, and phospholipase A2 [128]. Several studies also confirmed the inhibition of such enzymes by Cerrado plants, namely, *B. argyrophylla* [63], *E. dysenterica* [74], and *A. muricata* [59]. The study conducted by Pereira et al. on the species *A. crassiflora*, which is rich in chlorogenic acid, (epi)catechin, procyanidins, caffeoyl-hexosides, quercetin-glucosides, and kaempferol, suggested an antiobesogenic capacity of the isolated alkaloid compound stephalagine. Stephalagine demonstrated inhibitory activity in vitro against pancreatic lipase with low cytotoxicity [58].

### 3.2. Anti-Inflammatory Effect

Chronic inflammation in obesity and its comorbidities can also be treated by MPs, especially those native from the Cerrado. The phytochemical compounds, phenols, and flavonoids present in *Xylopia aromatica* were shown to reduce inflammation in overweight male BALB/c mice. The plant extract composed of flavonoids, namely rutin and caffeic and chlorogenic acid, was able to reduce glycemia during an oral glucose tolerance test, improve insulin sensitivity, attenuate inflammation and hepatic steatosis, and even showed potential in the release of anti-inflammatory cytokines, namely IL-4 and IL-13 in the liver and AT [123]. *Pyrostegia venusta* also exhibited anti-inflammatory effects in BALB/c mice induced with a high-carbohydrate-refined diet. Treatment with the extract of the plant showed an increase in IL-10 and a decrease in TNF-α, IL-6, IL-4, and IL-13 levels in AT. In the liver, the treatment decreased the TNF-α level. The plant extract was also able to reduce adipocyte area and adiposity, as well as to improve the metabolic profile of the animals [129].

In C57Bl/6 mice with a high-fat/high-sucrose diet, the reversal of the inflammatory condition was also achieved through *E. dysenterica* treatment, popularly known as cagaita. The plant extract is rich in polyphenols (total polyphenols, proanthocyanidins, ellagitannins, quercetin derivatives, kaempferol derivatives, and free ellagic acid) and was able to reduce NF-κB levels, protecting the animals from dyslipidemia and glucose alterations and reducing gluconeogenesis and hepatic inflammation [74,130]. Finally, the study conducted by Napolitano et al. demonstrated the anti-inflammatory effect of three other Cerrado plants, namely *Serjania lethalis*, *Cupania vernalis*, and *Casearia sylvestris*, through the measurement of nitric oxide released by macrophages [131].

### 3.3. Adipocyte Differentiation, Adipogenesis, and Browning

PPARs are members of a large family of nuclear receptors activated by fatty acids and their derivatives, regulating the metabolism of lipids and carbohydrates. They are predominantly expressed in the digestive tract, liver, and cardiac muscle and are involved in lipid catabolism. PPAR activators are widely used in the treatment of dyslipidemia, such as the pharmacological class of fibrates or Glitazones [34]. PPARs have also been described as a key transcription factor involved in adipogenesis [132], and they contribute to the reduction of inflammation and the transcriptional regulation of proinflammatory factors [133,134]. Another plant native to the Brazilian Cerrado, *Hippeastrum stapfianum*, showed a selective activation of PPAR-α and PPAR-γ and exhibited antioxidant effects. Although the focus of the study was not specifically metabolic diseases, such results suggest a possible effect on obesity [135]. Lastly, the study by Duarte et al. highlighted the properties of wild passion fruit (*Passiflora setacea*), which is rich in polyphenols. In individual male volunteers treated with *P. setacea* pulp, a significant reduction in insulin levels, HOMA IR, and IL-6 was observed. The plasma from the same individuals was used as a treatment in BV-2 microglial cells, and as a result, higher levels of PPAR-γ were observed in the cells, indicating an anti-inflammatory and antidiabetic effect [92].

There are no reports in the literature regarding the modulation of proteins such as C/EBP, FAS, and SREBP induced by Cerrado plants. However, some studies have shown a relationship between the presence of polyphenols and the increase in these proteins [136]. In addition, studies have also found an anti-obesity effect of catechins, which have been shown to reduce fat droplets in 3T3-L1 cells and the expression of transcription factors related to adipogenesis, such as SREBP-1c, C/EBP α, and PPAR γ [137]. An alternative to reduce obesity is the process of converting white adipocytes into brown adipocytes, known as browning. In brown adipocytes, PGC-1α promotes lipid oxidation and releases heat within the mitochondrial electron transport chain through the uncoupling protein 1 (UCP-1), a process called adaptive thermogenesis [138,139]. Although there are currently no studies relating Cerrado phytochemical compounds to browning, some reports observed this effect in plants rich in compounds such as those found in Cerrado plants, especially the great group of phenolic compounds and terpenes. They have been associated with the browning effect, adaptive thermogenesis, and the activation of target molecules such as PGC-1α, PPARγ, SIRT1, and AMPK [140]. This further justifies the investigation of these plants to determine their potential for future obesity treatment.

### 3.4. Modulation of Neuroendocrine Mechanisms 

Appetite reduction is considered the primary approach for regulating and restoring body weight, being regulated by more than 40 hormones, enzymes, neuropeptides, and their respective receptors. Phytochemical compounds that have appetite-inhibiting activity include epigallocatechin, steroid glycosides, and saponins. It is believed that a significant portion of appetite inhibition occurs through signals from the brain–gastrointestinal system axis [39]. The hormones involved in this mechanism can also have their expression increased by medicinal plants. Although studies on this subject are still scarce, considering the main effect of body weight control, we can suggest that this mechanism of action may occur with the Baru nut, which has shown significant effects in improving satiety and in reducing body mass in adults [141,142]. Furthermore, the fruits of *Solanum lycocarpum*, known by the population as ‘fruta-de-lobo’, are rich in polysaccharides that promote delayed gastric emptying and induce changes in the endocrine system, affecting gastrointestinal hormones and consequently improving glycemic and lipid profiles [143]. Additionally, Rocha and colleagues conducted a study with various native Brazilian nuts, including cashew nuts (*Anacardium occidentale*—native to the Cerrado region), and they found that, although not altering leptin levels, their consumption for a period of 8 weeks showed potential in reducing ghrelin levels and consequently in decreasing appetite and cardiometabolic risk [144]. Another study conducted by Torres et al. evaluated the metabolic effect of pequi (*C. brasiliense*) administered to rats for 21 days. The oil from pequi kernels demonstrated the ability to reduce serum leptin levels, IL-6, leukotrienes 4 and 5, and TNF receptor. The treatment also showed a reduction in liver lesions and biochemical markers of liver function, AST and ALT, and increased serum concentrations of HDL. In addition, it exhibited antioxidant effects by increasing the expression of enzymes such as glutathione peroxidase and reductase [145].

**Table 2 ijms-24-15588-t002:** The role of medicinal plants in regulating inflammation, adipogenesis, browning, endocrine mechanisms and enzyme activity.

Therapeutic Properties	Medicinal Plant	Experimental Condition	Refs.	Medicinal Plant	ExperimentalCondition	Model	Treatment/Dose	Refs.
In Vitro				In Vivo				
Antilipidemic effect	-	-	-	*Casearia sylvestris*	High fat diet	Swiss and C57BL/6 LDLr-null mice	250 and 500 mg/kg extract	[124]
	-	-	-	*Eugenia dysenterica*	High-fat high-sucrose diet	C57BL/6J mice	7 and 14 mg gallic acid equivalent of extract/kg	[74]
Enzymatic inhibition	*Kielmeyera coriacea*	α-amylase, α-glucosidase, and pancreatic lipase inhibition, antioxidant and antiglycation assays	[81]	-	-	-	-	-
	*Banisteriopsis argyrophylla*	α-amylase, α-glucosidase, and pancreatic lipase inhibition, antioxidant and antiglycation assays	[63]	-	-	-	-	-
	*Annona muricata*	α-amylase, α-glucosidase, and pancreatic lipase inhibition, antioxidant, antiglycation assays and cytotoxic assays	[59]	-	-	-	-	-
	*Annona crassiflora*	Lipase inhibition and cytotoxic assay	[58]	-	-	-	-	-
Anti-inflammatory effect	*Serjania lethalis*, *Cupania vernalis*, *Casearia sylvestris*	Determination of nitric oxide production and cytotoxicity assay	[131]	*Xylopia aromatica*	High carbohydrate	BALB/c mice	50, 100 and 200 mg/kg extract	[123]
	-	-	-	*Pyrostegia venusta*	High-carbohydrate-refined diet	BALB/c	300 mg/kg extract	[129]
	-	-	-	* Eugenia dysenterica *	High-fat high-sucrose diet	C57BL/6J mice	7 and 14 mg gallic acid equivalent of extract/kg	[74,130]
Adipocyte differentiation, adipogenesis, and browning	* Hippeastrum stapfianum *	Activation of PPAR-α, PPAR-γ and antioxidant assays	[135]	*Davilla elliptica*	High-lard/high-sugar diet	Swiss mice	0.26 mg/kg extract	[125]
	-	-	-	*Passiflora setacea*	Clinical trial	Overweight male volunteers and BV-2 microglial cells	50 g, 150 g of pulp in two phases (humans), phenolic mebolites (cells)	[92]
Modulation of neuroendocrine mechanisms	-	-	-	*Anacardium occidentale*	Clinical trial	Women at cardiometabolic risk	15 g of Brazil nuts + 30 g of cashew nuts	[144]
	-	-	-	*C* *aryocar* *. brasiliense*	Liver injury induction with carbon tetrachloride	Wistar rats	3 or 6 mL/kg of almond oil	[145]

## 4. Type 2 Diabetes, Oxidative Stress and Glycation

Metabolic syndrome (MetS) is a common metabolic consequence of overweight and obesity and is cluster of metabolic abnormalities characterized by the simultaneous presence of the following risk factors: (I) abdominal obesity, (II) T2D, (III) hyperinsulinemia and/or glucose intolerance, (IV) dyslipidemia, and (V) hypertension [19], which can be either genetically predisposed or acquired throughout life due to certain habits. Typically, T2D manifests within the framework of metabolic syndrome, which is characterized by concurrent conditions such as obesity, hyperlipidemia, hypertension, and coagulation disorders. These interrelated factors significantly contribute to the development of CVD [146], the primary global cause of mortality [147]. CVD encompasses a wide range of conditions, including cerebrovascular disease, peripheral artery disease, cardiomyopathy, coronary heart disease, arrhythmias, and sudden death [148]. It is crucial to note that diabetic individuals face a significantly elevated risk, ranging from 150% to 400%, of experiencing stroke episodes [146].

Assuming that there is a vicious cycle among the three mentioned conditions, MP with antioxidant potential will likely show an improvement in common metabolic conditions. Given the antioxidant effect of plants from the Cerrado biome, as described by several studies already mentioned using extracts rich in compounds with phenolic structure, many of them also have antidiabetic effects [149]. Table 3 summarizes the data found for plants showing their action on the regulation of oxidative stress, T2D and glycation.

### 4.1. Antidiabetic Effects of Cerrado Plants

Several drugs used in the treatment of non-insulin-dependent diabetes slow down the digestion of starch by suppressing the activity of starch-hydrolyzing enzymes such as pancreatic α-amylase and intestinal α-glucosidase [60]. Although medications such as acarbose, voglibose, miglitol, and other starch blockers have been successfully used to manage postprandial hyperglycemia in T2D, their associated side effects have sparked interest in alternative therapies. Recent research suggests that plant-derived bioactive phytonutrients, such as alkaloids, cardiac glucosides, flavonoids, tannins, phenols, and steorids, with antioxidant properties can function as inhibitors of pancreatic α-amylase and intestinal α-glucosidase activities [150]. This inhibitory effect, in the context of phytotherapy, is attributed to the redox potential of their hydroxyl groups, which serve as effective enzyme inhibitors [151]. Besides enzymatic inhibition, several studies have also demonstrated the antidiabetic effects of Cerrado plants. For instance, the leaves of *Annona muricata*, used in ethnopharmacology for the treatment of T2D and its complications, were evaluated by Justino et al. regarding their in vitro antidiabetic effect. They determined the inhibition of lipidic and glycosidic hydrolases such as α-amylase, α-glucosidase, and pancreatic lipase, associated with antioxidant and antiglycation effects performed through DPPH, ORAC, FRAP assays, the reduction of Fe^2+^, the reduction of lipid peroxidation, and albumin and arginine glycation. By preparing various fractions of the leaf extract, the authors confirmed the presence of bioactive compounds described as antioxidants, such as quercetin and caffeic acid. They demonstrated the antioxidant effect of *A. muricata* in the conducted tests and the reduction of albumin glycation in the assays with methylglyoxal. The authors associate the phenolic group and the presence of hydroxyl groups with the effects observed [59]. Another in vitro study conducted with *A. muricata* by Olasehinde et al. also revealed and associated the plant’s antidiabetic and antiglycation potentials with its phytochemical constitution (phenols, flavonoids, saponins, tannins, steroids, and alkaloids) [60]. Finally, Pinto et al. showed the antidiabetic effect of this same plant but now analyzed the oil from the seeds in a model of streptozotocin (STZ)-induced type 1 diabetes (T1D) in BALB/c mice and in erythrocytes of diabetic patients. The oil from *A. muricata* seeds was able to preserve pancreatic islets and hepatic tissue and improve the patients’ anti-inflammatory profile with higher levels of IL-4 and 10 and lower levels of IFN-γ in the patients’ blood [152].

Additionally, in the study conducted by Florence et al. carried out in STZ-induced rats treated for 28 days, a single administration reduced blood glucose levels from 58.22% to 75%, depending on the dose. The extract was also able to reduce serum creatinine, MDA levels, AST, ALT, and LDL cholesterol and restore triglyceride and total cholesterol levels, as well as the antioxidant enzymes superoxide dismutase (SOD) and catalase (CAT) [153]. Another plant with demonstrated antidiabetic potential is Acrocomia aculeata. Besides acting as antioxidant in several experimental models, the aqueous extract of its leaves was able to reduce blood glucose and triglyceride levels in non-obese type 2 diabetic rats (Goto-Kakizaki) after 30 days’ treatment, as well as increasing proteins involved in glucose and lipid metabolism, such as GLUT-4, PPARγ, AMPK, and IR. Regarding the redox balance, the authors observed increased cell viability and higher catalase levels in 3T3-L1 preadipocytes exposed to hydrogen peroxide [2]. The extract also improved vascular function in diabetic animals and exhibited an important antioxidant effect on dermal microvascular cells, HMVec-D. The authors also associate the therapeutic effects with the phenolic composition of the extract (gallic, vanillic, ferulic, and caffeic acids), although other metabolites may be produced in vivo after the extract administration to the animals [12]. Another study was conducted with the same plant but using the kernel oil. It also showed an improvement in metabolic parameters, such as the reduction of hyperglycemia, increased insulin secretion, decreased insulin resistance, and enhanced function and number of pancreatic β-cells in both diabetic and non-diabetic rats. Similar to the study conducted with the leaf extract, kernel oil was able to improve the lipid profile and promoted better liver function [154]. Moreover, Da Silva’s work performed on the pulp of the same plant demonstrated a positive effect on the glycemic metabolism of rats and an antioxidant potential, without showing evidence of cytotoxicity [155]. These data contribute to an understanding of the extensive therapeutic potential of just one Cerrado plant and its various parts.

Other studies have also demonstrated the relationship between chemical composition, antioxidant effects, and metabolic improvement. A study conducted with Banisteriopsis argyrophylla showed the presence of phytochemical compounds such as glycosylated flavonoids, catechins, procyanidins, kaempferol, and glycosides, as well as its antioxidant activity by DPPH and ORAC, and the inhibition of non-enzymatic glycation, α-amylase, α-glucosidase, and lipase [63]. Jatobá-do-cerrado (Hymenaea stignocarpa Mart.) presented similar data regarding the inhibition of starch hydrolases with phytochemicals such as caffeic acid, kaempferol, quercetin-3-rutinoside, and quercetin-3-rhamnoside. Silva et al. proposed the substitution of a part of wheat flour by jatobá flour in bread and observed a less dose-dependent glycemic response. They confirmed the synergistic effect of dietary fibers and the phenolic compounds of Jatobá on glucose metabolism [79]. Eugenia dysenterica was also evaluated for its potential inhibitory effects on hydrolases, as well as its antioxidant and antiglycation properties. The authors found quercetin and gallic acid in its composition and confirmed the antidiabetic, antiglycation, and antioxidant effects of the plant [75]. Another interesting study conducted with Pequi (Caryocar brasiliense) demonstrated the presence of phenolic acids and tannins as constituents, including corilagin and geraniin, and linked them with the inhibition of insulin resistance-related factors, such as TNF-alpha and IL-1B, and inhibitory effects over α-glucosidase. In a starch tolerance test in mice, Pequi was able to reduce blood glucose levels [67]. Terminalia phaeocarpa also demonstrated enzymatic inhibition, while reducing in vivo blood glucose and TNF-alpha and IL-1B levels [156].

Other studies demonstrated the vast potential of the Cerrado plants Eugenia florida, Alibertia edulis, and Terminalia phaeocarpa in diabetic animal models, also with phytochemical composition mainly composed of phenols. They showed improvements in biochemical profiles, including higher levels of HDL cholesterol, improved liver function through AST and ALT markers, reduced uric acid, and reduced renal damage. They also presented a better redox condition, with lower levels of malondialdehyde and higher levels of glutathione (GSH) [55,157,158].

Regarding cashew (*Anacardium othonianum*), it is rich in vitamin C and was able to improve glucose levels during the glucose tolerance test, serum lipid levels, and systolic blood pressure in control women [159]. In a study conducted by Talpo et al. in C57Bl/6J mice, *Siolmatra brasiliensis* was able to reduce insulin resistance and lower serum cholesterol levels. Animals induced with HFD diet and treated with the extract showed a reduction in body weight and caloric intake, besides a 17% reduction in fasting blood glucose. Treatment also decreased biochemical markers related to glycation and lipid peroxidation, such as AGEs and TBARs, and increased antioxidant defenses such as CAT and SOD [160].

Finally, based on ethnopharmacological knowledge, cagaita (*E. dysenterica*) was studied for its antidiabetic potential in dysglycaemic female volunteers with metabolic syndrome. Araujo et al. showed that the treatment with cagaita juice was able to reduce 53% of the incremental area under the curve (iAUC) of glucose, 38% of insulin, 78% of glucose-dependent insulinotropic polypeptide (GIPb), and 58% of C-peptide, after juice consumption and bread intake, compared to control water. Glucagon-like peptide-1 (GLP-1) and glucagon values were not affected by the treatment with cagaita [161].

**Table 3 ijms-24-15588-t003:** Medicinal plants with potential therapeutic effects for treating T2D, oxidative stress, and glycation.

Therapeutic Properties	Medicinal Plant	Experimental Condition	Refs.	Medicinal Plant	Experimental Condition	Model	Treatment/ Dose	Refs.
	In Vitro			In Vivo/Clinical Studies				
Enzymatic inhibition and antidiabetic effects	*Annona muricata*	α-amylase, α-glucosidase, and pancreatic lipase, associated with antioxidant and antiglycation assays	[59]	*Annona muricata*	Mice—streptozotocin (STZ)-induced T1D	Male BALB/c mice and in erythrocytes of diabetic patients	1.0 mg/kg seed oil	[152]
	*Annona muricata*	antidiabetic, and antiglycation potentials	[60]	*Annona muricata*	STZ-induced T2D	Male wistar rats	100 mg/kg or 200 mg/kg extract	[153]
	*Acrocomia aculeata*	Antioxidant and cytotoxic assays	[2]	*Acrocomia aculeata*	Normal and non-obese T2D rats	Male wistar and Goto-Kakizaki rats	200 mg/kg extract	[12]
	-	-	-	*Acrocomia aculeata*	STZ- and low HFD induction	Male wistar rats	40 or 160 g of kernel oil	[154]
	Banisteriopsis argyrophylla	Inhibition of α-amylase, α-glucosidase, and lipase and antioxidant assays	[63]	*Acrocomia aculeata*	Normal, STZ and frutose-induced diet	Male wistar rats	3, 30 or 300 mg/kg pulp oil	[155]
	Eugenia dysenterica	Inhibitory effects on hydrolases, antioxidant and antiglycation properties	[75]	*Hymenaea stignocarpa*	Normal and healthy	Women	Replacement of normal flour with pulp flour at 10, 20 and 30%	[79]
	-	-	-	*Caryocar* *brasiliense*	Normal rats	Male swiss mice and THP-1, CCL-13 and CR-1458 cell lines	100 mg/kg extract and fractions	[67]
	-	-	-	*Terminalia phaeocarpa*	Normal rats	Male swiss mice and THP-1 cell line	100 mg/kg extract and fractions	[156]
	-	-	-	*Eugenia florida*	Normal and Alloxan-induced	Male Wistar rats	200 mg/kg extract	[43]
	-	-	-	*Alibertia edulis*	Normal and HDF-induced	Male swiss mice	200 and 400 mg/kg extract	[53]
	-	-	-	*Anacardium othonianum*	Normal and healthy	Women	400 mL juice	[159]
	-	-	-	*Siolmatra brasiliensis*	Normal and HDF-induce	Male C57Bl/6J mice	125 or 250 mg/kg extract	[160]
	-	-	-	*Eugenia dysenterica*	Metabolic syndrome	Woman	300 mL juice	[161]
Redox imbalance and antioxidant potential	* Mauritia flexuosa *	Free radical scavenging	[85]	*Acrocomia aculeata*	Oxidative stress induction	*C. elegans*	500–1000 μg/mL	[2]
	* Mauritia flexuosa *	Free radical and hydroxyl scavenging, and reducing iron	[82]	*Acrocomia aculeata*	Normal and non-obese T2D rats	Wistar and Goto-Kakizaki rats	200 mg/kg	[12]
	*Annona crassiflora*	ABTS free radicals capture	[162]	-	-	-	-	-
	*Solanum lycocarpum*	DPPH, FRAP and ORAC techniques	[99,163]	-	-	-	-	-
	*Dipteryx alata*	Antioxidant profile with antiproliferative activity	[69]	-	-	-	-	-

### 4.2. The Redox Imbalance and Antioxidant Potential of Phytochemical Compounds from the Cerrado Plants

Oxidative stress is an important factor transversal to various conditions such as cancer, premature aging, CVD, and metabolic disorders such as T2D and obesity [164]. ROS formation is a consequence of secondary glucose metabolism and the upregulation of alternative pathways that lead to glucotoxicity [165,166,167,168,169,170]. Oxidative damage plays a crucial role in neuronal damage and microvascular and cardiovascular complications of diabetes, such as nephropathy, retinopathy, and neuropathy, mainly due to glucotoxicity [12,171]. In hyperglycaemia, the glycolytic pathway, Krebs cycle, and mitochondrial electron transport chain (METC) are upregulated. This leads to higher levels of ATP and hyperpolarization of METC, promoting electron “escape” and partial oxygen reduction, resulting in higher levels of ROS, especially O_2_^•−^ [172]. Additionally, glycolysis alternative pathways are activated, including protein kinase C (PKC), aldose reductase (polyol pathway), increased hexosamine flow, and advanced glycation end product (AGEs) formation [173]. All these alternative pathways are associated with ROS increase and are linked to the development of oxidative stress and T2D complications. Clinical indicators of oxidative stress include 4-hydroxy-2-nonenal (HNE) [169,170], 8-hydroxy-2-deoxyguanosine (8-OHdG) [165], and heme oxygenase-1 (HMOX1) [166,167]. The antioxidant enzymatic system is formed by enzymes such as SOD, glutathione peroxidase (GPx), and CAT [174]. The non-enzymatic system includes substances such as uric acid, lipoic acid, melatonin, bilirubin, and GSH. Exogenous antioxidants can be divided into two groups: natural and synthetic. Natural antioxidants are represented by phenolic compounds, carotenoids, flavonoids, and vitamins E, A, and C, among others [175]. They neutralize ROS from metabolism and exogenous sources, thereby preventing macromolecule damage. Synthetic antioxidants are more effective, stable, and cheap than natural ones but are associated with more side effects [176].

Several recent studies have reported an association between oxidative stress and AT dysfunction, namely elevated levels of FFA, dysregulation in adipokine production, and a reduced expression of antioxidant enzymes [177,178]. Thus, improving redox imbalance may attenuate the dysregulation of white AT and improve its function. The development of T2D is also closely related to the redox condition, being associated with reduced insulin biosynthesis and secretion [179]. The reduction of insulin biosynthesis in pancreatic β cells is further exacerbated by oxidative stress, which hampers the activity of the insulin gene’s promoter and transcription factors pancreatic duodenal homeobox-1 (PDX-1) and MAF bZIP transcription factor A (MafA) [180,181]. Besides decreasing the expression of insulin mRNA, the development, differentiation, maintenance, and regeneration of β cells are also compromised [179]. Consequently, the combination of hyperglycemia and oxidative stress leads to a decrease in insulin biosynthesis and secretion, thereby perpetuating a vicious cycle of sustained hyperglycemia. AGEs result from glucose auto-oxidation or the interaction of precursor metabolites (namely glyoxal and methylglyoxal, which are metabolic byproducts of glycolysis) with proteins and amino acids. Arginine is particularly important when assessing protein glycation caused by reactive carbonyl compounds. The inhibition of theses reactions, as well as the subsequent oxidative products, is of the utmost importance for the prevention and reduction of T2D complications, including the impairment of vessel function [182] and insulin resistance [183]. 

Natural compounds can act as antioxidants in various ways. Besides neutralizing ROS, their action includes the repair of ROS-induced injury, increasing the synthesis of antioxidant enzymes, and reducing oxidative damage [184,185]. One example of a Cerrado plant that is a source of antioxidant agents is *A. aculeata*, which has shown significant effects in capturing free radicals and reducing the oxidation of macromolecules (DNA, cell membrane lipids, and proteins). Extracts prepared from its leaves have been shown to be rich in phenolic compounds, tannins, and saponins (mainly gallic acid, vanillic acid, caffeic acid, ferulic acid, rutin, and quercetin) and to increase the survival of nematodes *Caenorhabiditis elegans* and normal cells Cos-7 exposed to oxidative damage. Furthermore, they have been found to increase the expression of protective transcription factors such as NRF2 and SIRT1 and antioxidant enzymes such as CAT, which were associated with the protective in vivo effects in diabetic rats [2,12].

The buriti palm (*Mauritia flexuosa* L.), a native palm tree of Latin America, especially from the Cerrado biome, is an important source of bioactive compounds such as carotenoids and polyphenols. In the analysis conducted by Leite et al., catechin, quercetin, and gallic acid were the major compounds in the fruit peel and pulp, demonstrating antioxidant effects in the scavenging of the ABTS free radical [85]. Candido et al. performed a comparative analysis between samples collected from two different Brazilian biomes, the Cerrado and the Amazon rainforest. Samples from the Cerrado exhibited a higher content of phenolic compounds and showed superior in vitro antioxidant activity. The buriti palm from the Cerrado displayed greater potential in capturing free radicals, reducing iron, and scavenging hydroxyl radicals, as confirmed by DPPH, ABTS, FRAP, and ORAC analyses, compared to the samples from the Amazon [82]. Other studies in the literature also confirmed the antioxidant effect of the buriti palm, associating this effect with its phenolic composition [83,84,86,87,89] and corroborating the previously mentioned antimicrobial activity data [84]. Furthermore, studies confirm its safety through antimutagenicity and micronucleus assays [88].

The study conducted by Rios de Souza et al. on five different species highlights the antioxidant potential of Marolo (*Annona crassiflora* Mart.), which showed a significant effect in capturing ABTS free radicals, a notable content of ascorbic acid, and a higher phenolic compound content, compared to the other four evaluated plants [162]. The study conducted with *Solanum lycocarpum* St. Hil., another native plant of the Brazilian Cerrado, linked the antioxidant effect determined in different extracted fractions of its fruits by DPPH and FRAP techniques. The chemical composition of this assay revealed the presence of phytosterols and terpenes [163]. The same plant was studied by Pereira et al., demonstrating once again its antioxidant potential, this time using the ORAC technique, in addition to the presence of phenolic compounds in its composition [99]. *Dipteryx alata* Vog. is another medicinal plant with therapeutic potential. It was analyzed by Oliveira-Alves et al., who associated its antioxidant profile with antiproliferative activity in human colon cancer cells HT29 and Caco-2. The extract rich in phenolic compounds such as gallic acid and its derivatives demonstrated its potential through the ORAC assay and hydroxyl radical scavenging capacity (HOSC) assay [69]. These are crucial studies that contribute to the characterization of the chemical composition and antioxidant profile of different species in the Cerrado biome. These plants are promising in reducing redox unbalance and thus in preventing the oxidative stress associated with metabolic disorders.

### 4.3. Limitations of the Study

In this review, our primary aim was to summarize the studies exclusively conducted on Cerrado plants. While there is a significant number of publications, many studies lack comprehensive descriptions of the plants’ phytochemical composition, and it is not yet possible to establish a composition–effect relationship. However, we chose not to exclude studies that did not present the full characterization of bioactive compounds, emphasizing the Cerrado’s extensive biodiversity even though it remains poorly documented. Methodological descriptions remain relatively underdeveloped, especially with regard to the study of mechanistic pathways related to bioactive compounds and their biological effects. In Figure 2, we summarize the possible mechanisms of action that may be related to the therapeutic potential of Cerrado plants for the treatment of obesity and its consequences.

Cerrado plants exhibit remarkable therapeutic potential, as proven in the course of our work. However, despite their promise, there are also few studies involving humans, and no reports were found of plants being evaluated in clinical studies for the development of new drugs. It is important to highlight that there is an incentive for the use of medicinal plants applied in phytotherapy, such as the National Policy on Medicinal Plants and Phytotherapy (PNPMF) of Brazil, which aims to guarantee the population’s safe access to medicinal plants and herbal medicines, promoting the responsible use of these natural resources. Despite these phytochemical compounds being strong candidates for the development of new allopathic drugs, extensive research is still needed.

Additionally, the low bioavailability of secondary metabolites, especially phenolic compounds and flavonoids, represents a significant challenge for medicinal plant research. The results obtained in in vitro studies are often not reflected in in vivo tests [186] due to characteristics such as low water solubility and instability in the gastrointestinal tract, in addition to rapid metabolism—most often hepatic—and low membrane penetration [187]. On the other hand, hepatic metabolism and the microbiota can contribute to increased therapeutic efficacy, producing more active metabolites, as we have already reported in another article published by our group [12]. In this context, two equally important aspects require further investigation: the identification of both inactive and more active metabolites, as well as the assessment of the therapeutic efficacy and reproducibility of data obtained in in vitro assays in in vivo models. Comprehensive studies are necessary to fully realize the potential of these plants.

## 5. Conclusions

The Brazilian mega-biodiversity is still a largely unexplored universe and a source of diverse bioactive compounds with promising pharmacological properties. These phytochemical compounds are potential candidates for the development and bioprospection of new therapeutic strategies that can be used to treat obesity and its metabolic sequelae. With all the reports described here, we have demonstrated the incredible potential that is available to be explored, with anti-inflammatory, antioxidant, anti-obesogenic, and antidiabetic properties, among others, which can contribute to the maintenance and restoration of metabolic health. Based on the evidence presented here, we aim to encourage the conduction and development of new research that allows the exploration of these still less-known plants, as well as emphasizing the need to raise nature consciousness towards biodiversity preservation while promoting scientific and technological development.

## Figures and Tables

**Figure 1 ijms-24-15588-f001:**
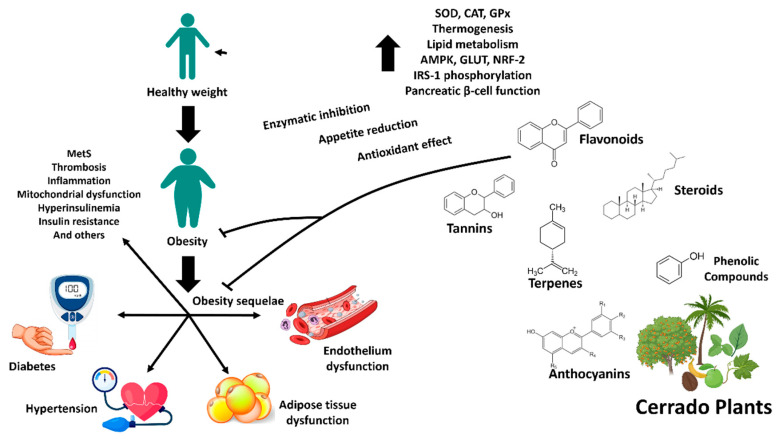
Main classes of molecules found in Cerrado plants and their potential therapeutic targets in obesity.

**Figure 2 ijms-24-15588-f002:**
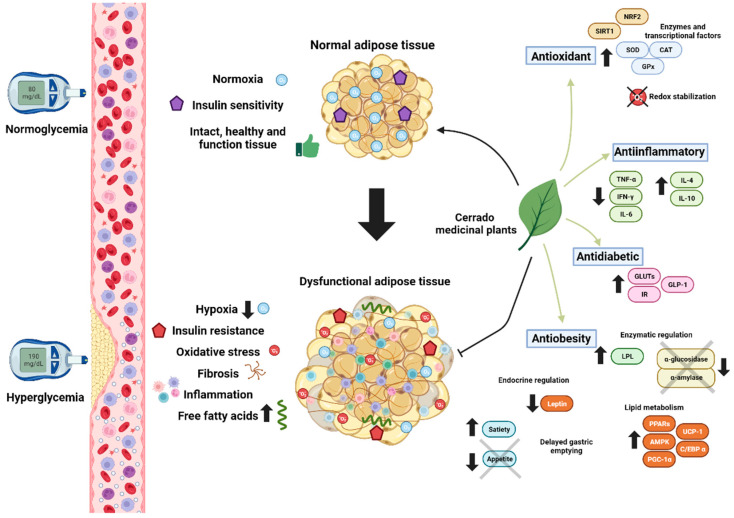
Summary of the main mechanisms involved in the pathophysiology of obesity and its metabolic complications, and the role of Cerrado plants in improving such mechanisms. The role of Cerrado plants in regulating several mechanisms is not yet know, as it is also not known which compounds are responsible for the therapeutic effects of the plants.

## Data Availability

Not applicable.

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
