# Peer review of "New Therapeutic Strategies for Obesity and Its Metabolic Sequelae: Brazilian Cerrado as a Unique Biome"

_ijms, 2023, doi:10.3390/ijms242115588_

Round 1
Reviewer 1 Report
Dear Author(s),
Thank you for your manuscript - it is extensive and interesting, but in the present form it can be misleading.
I am willing to provide you re-peer-review after this modification:
References you are mentioning as the evidence should be analyzed as per EBM and GRADE methodology. Please mention study designs and if the outcome analyzed was primary or secondary (which is only exploratory and under-powered); thus, evaluate the level and strength of evidence (within the text and tables).
After this important modification I can proceed with re-peer-reviewing; since, in the present form, to the best of my knowledge, article may be misleading if nor organized like that. Also consider making a systematic search strategy in order to obtain all the evidence out there to avoid the bias which now is most probably present.
Best regards, Reviewer
Only smaller modifications needed.
Author Response
The authors of the article entitled “New therapeutic strategies for obesity and its metabolic sequelae: Brazilian cerrado as a unique biome”, thank reviewers for their attention and care in evaluating our work. We do not highlight changes because all the text was revised and modified in order to improve the clarity and quality of the text. All the sections were revised, a new tables and a figure were included.
# reviewer 1:
We believe EBM or GRADE methodology is not the most appropriate for our work, since data is still very scarce in the area and fragmented, as mentioned in the text. Our objective is not to perform a systematic analysis of the effects of a specific plant, but the describe the current knowledge of all plants. Most of the studies don’t have the full chemical characterization, some of them use different parts of the same plant and use those preparations for different clinical applications. This makes it impossible to carry out a consistent meta-analysis. Our aim was to carry out a survey of existing publications on the topic and to organize this data to guide future studies. This is an exploratory review, as quantitative as possible and more in-depth data in terms of mechanistic and chemical description are still scarce and inadequate for an EB or GRADE-type assessment.
This type of approach is crucial for the structuring and basis of scientific knowledge regarding plants, and given the limited amount of data we have available so far, we believe it is the only way to compile the little information available. However, considering the comments of all reviewers, we made several adjustments to the text, including two more tables targeted at the biological effect and all the studies exploring the different approaches and plants. Our goal is to provide a document with critical and constructive information, considering the current limitations of available materials, paving the ground for future research.
Reviewer 2 Report
This review article provides useful information about the anti-obesity properties of Brazilian plants. However, the manuscript seems to be confused.
The author should separate the in vitro studies and the in vivo animal studies in the text and should include their main results in two separated tables.
- Moreover, clinical human studies should be described all together fter in vivo studies by cinstructing abather table only for them.
- A discussion section where the authors could compared the existing knolefgment is missing.
- Also, the authors should scrutinize the reported studies by reporting their potential strebgth and limitaions. A relevant section should be included at the end of the discussion section.
-The authors should also reoort the poor bioavailability of several plant-derived compounds such as flavonoids.
- Could these naturrally ocuring compounds mayy used as drug candidates in Drug Design process?
- A discussion about the dosage used to exert their benficial effects since overdose may lead to adverse side effects.
- Moderate english language editing is required.
Moderate english language editing is required.
Author Response
The authors of the article entitled “New therapeutic strategies for obesity and its metabolic sequelae: Brazilian cerrado as a unique biome”, thank reviewers for their attention and care in evaluating our work. We do not highlight changes because all the text was revised and modified in order to improve the clarity and quality of the text. All the sections were revised, a new tables and a figure were included.
#reviewer 2: The in vitro and in vivo (animal and human) studies were separated into sections and new tables were made. However, in relation to clinical studies, only 4 studies were found with plants from the Cerrado. For this reason, we chose to divide the tables into in vitro and in vivo studies only. The information regarding dosages was included in the new tables. Moreover, a section responding to their interesting notes was included in the text, named “limitations of the studies found”. The questioned grammatical analysis was carried out.
Reviewer 3 Report
Give examples for “unhealthy lifestyle habits” Line 47.
Do all secondary metabolites have effects on the angiotensin-converting enzyme (ACE1 and 2)? Line 93
Again, do all herbs and their derived compounds have the potential to control appetite, increase satiety, inhibit pancreatic lipase activity, ….? Line 101. Those generalized sentences make bad impression, misunderstanding and great contradiction.
Explain structural changes of flavonoids in the human body. Line 135
Do all terpenes have anticancer activity? Line 144
In table 1, the authors should include the part used of each medicinal plants (i.e., leaves, fruits, stems, etc.) and the type of extract whether alcoholic or aqueous or a fraction from the total extract. Also, they should include more specifications about the biological activities whether they were conducted in vivo or in vitro, the type of cancer investigated.
Some words and complete paragraphs were written in italic for no reason like Quercetin and gallic acid or underlined like Friedelin.
The references like [71,72,72–77] and [2,42,90,91,91–94] cited in text contain duplicated number.
In table 1, sometimes the mechanism of biological activity is mentioned while in other positions it is not provided.
What do you mean by free and total ellagic acid?
The authors should recheck that each plant listed in table 1 is 100% safe concluded from evidence based toxicological studies and provide the proper references for such allegation.
Section 3, the characters of AT dysfunction and the underlined mechanisms for treatment should be illustrated in a detailed figure.
What do you mean by rich in polyphenols (total polyphenols,.. Line 250
Rephrase the sentence in line 361.
What is the kind of extract mentioned in line 362?
Line 373, the cited reference MONTEIRO-ALFREDO is written in uppercase, while the relevant paragraph is written in italic.
Authors should explain the reasons behind the findings in line 492.
No comments
Author Response
The authors of the article entitled “New therapeutic strategies for obesity and its metabolic sequelae: Brazilian cerrado as a unique biome”, thank reviewers for their attention and care in evaluating our work. We do not highlight changes because all the text was revised and modified in order to improve the clarity and quality of the text. All the sections were revised, a new tables and a figure were included.
In response to reviewer 3:
-The definition of “unhealthy habits” was included in the text.
-The information about “angiotensin-converting enzyme” was removed from the text, as it is not exactly the focus of the work.
-Not all herbs and their derived compounds have the potential to control appetite, increase satiety, or inhibit pancreatic lipase activity -We do not claim that ALL plants control such mechanisms so we changed the text accordingly in line 313.
Explain structural changes of flavonoids in the human body. Line 135 - Flavonoids undergo structural changes in the body due to the process of metabolization and conjugation. Metabolites are generated from this process, often inactive or even more active than when compared to their first conformation. This condition can become a limiting factor regarding its biological potential, and precisely for this reason, this information was also added to the text.
Do all terpenes have anticancer activity? – Just like in the same section, where we mentioned the anticancer effect of flavonoids, once again, we mentioned it in a generic way, as the intention in this part of the manuscript is only to describe the great potential of this class of secondary metabolites so that in this way, readers can understand the wide range of possibilities that flavonoids and terpenes can present. We never attribute the therapeutic effect to one class or another. It is even completely understandable, and this type of description is widely used in reviews like this.
-In Table 1, the authors should include the part used of each medicinal plant (i.e., leaves, fruits, stems, etc.) and the type of extract whether alcoholic or aqueous or a fraction of the total extract. Also, they should include more specifications about the biological activities whether they were conducted in vivo or in vitro, the type of cancer investigated.
We include more information in Tables 2 and 3, which were designed for the biological effects and not for the species.
-Some words and complete paragraphs were written in italic for no reason like Quercetin and gallic acid or underlined like Friedelin.
Thank you for this note. Corrections have already been made to the text.
-The references like [71,72,72–77] and [2,42,90,91,91–94] cited in text contain duplicated number.
Thank you, this was corrected
-In Table 1, sometimes the mechanism of biological activity is mentioned while in other positions it is not provided.
The information presented in the table is the one provided in the referenced articles. Not all of them are in fact complete about the mechanism of action and chemical characterization. For this reason, as suggested by another reviewer, a topic on limitations of the work was added at the end of the discussion.
-What do you mean by free and total ellagic acid?
The main difference between free ellagic acid and ellagic acid is in their chemical form and availability. Free ellagic acid is found in its isolated form and is readily absorbed by the body, while bound ellagic acid is associated with other molecules and needs to be broken down before being absorbed. Both forms have potential health benefits and are found in foods, but their absorption varies.
-The authors should recheck that each plant listed in table 1 is 100% safe concluded from evidence based toxicological studies and provide the proper references for such allegation.
The purpose of our work is not to actually analyze the toxicological profile of Cerrado plants that have biological properties. This request unfortunately cannot be met also because not all plants have a described toxicological profile. Manuscripts in this area of research often end up providing little evidence or studies that are not completely conclusive. Precisely for this reason, the topic in the manuscript that describes the need for toxicological analysis was rewritten and mentions the need for more studies in the area (line 263).
-What do you mean by rich in polyphenols (total polyphenols,.. Line 250
dysenterica extract is rich in polyphenols, that is, its chemical constitution is mostly made up of polyphenols, which are exemplified in parentheses.
- Line 373, the cited reference MONTEIRO-ALFREDO is written in uppercase, while the relevant paragraph is written in italic.
The error has been fixed. We appreciate your attention.
Rephrase the sentence in line 361.
What is the kind of extract mentioned in line 362?
Authors should explain the reasons behind the findings in line 492
-In relation to these requests, we kindly ask that reviewer 3 describe in full which phrases he is referring to. When sending the first version of the manuscript to the journal, our document did not have line numbers so we could not identify the sentences. However, most of the manuscript was improved, and we hope these sentences were also improved
Round 2
Reviewer 2 Report
The authors have added almost all of my suggestions and their manuscript has significantly been improved.
Moderate English language editing is required since there are several English mispellings and syntax/grammar mistakes
Moderate English language editing is required since there are several English mispellings and syntax/grammar mistakes
Reviewer 3 Report
Delete the TOC.
Language is fine.